# "Learn to Conserve Your Passion and Care": Exploring the Emotional Labor of Special-Post Teachers in Rural China

Jianjian Wu [1] and Huan Song [2,3,*]

1   Faculty of Education, Beijing Normal University, Beijing 100875, China
2   Center for Teacher Education Research, Beijing Normal University, Beijing 100875, China
3   Taihang Branch of Center for Teacher Education Research of Beijing Normal University, Xingtai University, Xingtai 054001, China
*   Correspondence: songhuan@bnu.edu.cn

**Abstract:** In recent decades, the growing trend of post-structuralist research on teacher emotional labor has offered a discursive lens to elucidate rural teachers' identities and their teaching practices. To date, however, few studies have explored the emotional labor of special-post teachers in rural China. Through a post-structuralist framework, this study aimed to explore the emotional labor of special-post teachers. Ethnographic qualitative data from a rural primary school in northern China showed that special-post teachers experienced various emotional conflicts embedded in multiple discourses. As teaching experience increases, special-post teachers obtain agentive emotional and practical responses to lighten their negative emotional burden for work. The findings suggested that the role overload and conflicts of special-post teachers were especially prominent in the social context of the urban–rural dichotomy. Emotional reflexivity and vulnerability of special-post teachers in their identity construction as educator, professional-service-provider, and also passer-by were also discussed.

**Keywords:** emotional labor; special-post teachers; rural schools; teacher identity; mainland China

## 1. Introduction

Considerable evidence from many countries supports that teachers in rural and remote regions are more likely to leave the teaching profession than those in urban or metropolitan schools, with higher teacher turnover rates [1–4]. In particular, it is quite a challenge for schools in rural regions or low-income communities to retain young teachers [5–7]. Many countries have enacted policies to try to alleviate the rural teacher shortage problem. In China, for instance, the central government has implemented the Special-Post Teacher Recruitment Program (abbreviated as SPTRP) since 2006, aiming to supplement teacher shortages in impoverished regions by adding college graduates to teach in rural schools for 3–5 years contractually [8]. Scholars found that working conditions and school characteristics, such as low salary levels, heavy workloads, lack of professional development resources, insufficient administrative support, and individual factors such as gender, might be reasons for rural teacher turnover [9–11].

In recent decades, scholars have highlighted that the emotional nature of the teaching profession could explain why teachers leave their careers so early [12,13]. The significance of teachers' emotions to both students and teachers themselves has been demonstrated [14,15]. Teaching is publicly accepted as one of those professions with a high level of emotional demands, which requires emotional labor [16,17]. Hochischild defined emotional labor as an individual emotional management behavior that intentionally induces or suppresses inner emotions in line with particular norms or rules, including two approaches, namely, surface acting and deep acting [18]. Emotional labor is regarded as one of the key factors of teacher burnout [19], which might cause teacher turnover intention to emerge. A large number of previous studies has explored the factors and consequences of teachers' emotional

labor from a psycho-dynamic perspective and a social constructionist perspective [20–24]. However, neither perspectives of emotional labor thoroughly explained how teachers' true feelings are shaped and reshaped by external structures when leaving school, since teachers' decision making processes were deeply linked with their perceptions and feelings about themselves as teachers and teaching profession. Furthermore, their sense-making processes are influenced by past daily interactions, dialogues, and experiences. In other words, the "self" of a teacher is supposed to be concerned to understand teachers and their teaching.

The increasing interest in post-structuralist emotion research has provided a discursive lens to elucidate rural teachers' identities and their practices. From a post-structuralist perspective, emotion and the self are inseparable [25]. What is taken as the core is the relationship between the institutional regulation of emotions and teachers' agentive responses and practices [26]. Research on teachers' emotional labor from a post-structuralist perspective not only reveals the cultural values and rules expected of teachers but also interrogates how teachers struggle in the complex process of becoming [27]. Although a handful of scholars have explored teacher emotional labor in rural areas or low-income communities [28–30], few have explored the interplay of emotional labor and the identity of rural teachers from a post-structuralist perspective.

As a unique group of rural teachers in mainland China, special-post teachers refer to those college graduates who participate in the Special-Post Teacher Recruitment Program (abbreviated as SPTRP) [31]. The SPTRP is the most impactful education policy for rural teacher supplementation [32], as well as a typical alternative hiring policy in the context of China aiming to recruit teachers for Chinese rural schools located in impoverished regions [33]. However, evidence has demonstrated that a considerable proportion of special-post teachers hoped or chose to leave after the service period [34,35]. In some regions, the attrition rate of special-post teachers was found to be relatively high [36]. Therefore, how to retain special-post teachers has become one of the core issues for developing Chinese rural education.

Evidence has suggested that the problem of emotional exhaustion among rural teachers in China is prominent [37]. Emotions significantly affected the commitment and identity of special-post teachers to continue staying in rural areas [38]. However, little is known about what emotional struggles special-post teachers experience and how they respond emotionally and practically in the discourses in which their identities are formed. Drawing on ethnographic qualitative data, this study aimed to explore the emotional labor of special-post teachers in Chinese rural areas from a post-structuralist perspective.

## 2. Literature Review

### 2.1. Understanding Emotional Labor from a Post-Structuralist View

In a post-structuralist view, emotion is not private nor completely socially constructed. Emotion, as one of the bodily and psychological attributes that are components of the self, is regarded as a discursive practice that is historically constituted and managed by social norms and disciplines [39,40]. According to Foucault, there is no distinction between a true or a fake self. All aspects of a person's self are subject to the outcomes of discursive practices, including emotions [40]. Through various practices and discourses, a set of norms or disciplines spreads [41]. Power relations are highlighted in understanding discourse. Foucault points out that power is not a possession that is acquired, seized, or shared. It is "the name that one attributes to a complex strategical situation in a particular society", which is dispersed in pervasive unequal and dynamic relations [42]. Therefore, emotions, as the constitution of the self, are embedded in power relations and are continuously fabricated within the social and cultural structure [43].

Emotional rules are crucial to understanding emotional labor. *Emotional rules* are developed from *feeling rules* [44], referring to the principles and rules that regulate and police individuals' emotions, especially in certain workplaces or professions. According to Zembylas (2005), emotional rules "are defined in relation to the social and individual self, the interplay of forces and desires, and the social structure of feelings, including the mutual

construction of the individual and society", continuously negotiated and constructed [40]. Embedded emotions are judged appropriate or inappropriate, permitted or not permitted in specific contexts by emotional rules that emotional discourse established [45], dissolving the clash between external feeling rules and inner authentic feelings. In this dynamic and discursive process, emotional labor is generated, historicized, and shaped through various discourses and practices. Hence, in the framework of post-structuralist approaches, there is no authentic self that has to violate itself to display inauthentic feelings when being demanded, since "the post-structuralist self is not an essential structure"; instead, it is discursively constituted and subjected to and by discourses of power [46]. Like all aspects of identity, emotional dissonance of the self is produced and embedded in and moderated by multiple and contradictory discourses interactionally [47]. Zembylas (2005) pointed out that emotional labor is a self-formation process in the tension among power relations within discourses [40].

Notably, although post-structuralist emotional labor is discursively embedded in structural discourses, there is still space for agency. According to Foucault (1978), where there is power, there is resistance, indicating that agency is embedded in the tension of domination and resistance [42]. The body is actually a significant location for resisting power and discourse conflicts. Self-identity is a continuity with awareness derived from the experiences of participation in multiple discourses and being reflexively constructed and organized [48]. The self, thus, is an agent with feeling and thinking and capability to act, reflect, and innovate. When placed in various emotional rules, individuals are capable of having emotional responses reflectively and actively. Emotional labor in a post-structuralist approach involves self-reflection on emotions and the possibility of making changes to negotiate with external power discourses, constantly revising emotional rules. This agentive aspect of emotional labor highlights the potential to signal challenges and promote change [26]. In this research, emotional labor is theorized as conflict or tension between the power relations of discourses in the special-post teachers' workplaces, in which agency is contained. In short, the resistance, compromise, and agentive responses of individual teachers in complicated emotional discourses provide a more abundant possibility for teacher emotional labor research [49].

### 2.2. Teachers as Emotional Laborers

Emotional labor is one of the common features of teachers' professional practice and an indispensable part of the teaching profession [50,51]. Teachers are recognized as emotional laborers since they are not only "acting out" their feelings superficially but also "consciously working oneself up into a state of actually experiencing the necessary feelings that are required to perform one's job well—be they feelings of outrage or enthusiasm, coolness or concern" [52]. Researchers have discussed the particularity of teachers' emotional labor. First, emotional management belongs to the moral principles of teaching [53]—teachers choose to be sensitive and empathic for their students, since caring for students is a reflection of teaching professionalism [54]. In addition, the outcomes of emotional labor in teaching are both negative and positive. Although emotional labor might cause teacher burnout or hurt teachers, the performance of emotional labor might also have negative impacts on teacher commitment to teach, satisfaction, and engagement [16,55,56]. Emotion management for teachers at work in interactions is regarded as crucial to the construction of their well-being [57]. Therefore, Day (2018) argued that teaching was undoubtedly emotional work, and in an ideal situation, good teaching means teachers actively engage themselves in understanding the student's emotions out of professional empathy [58]. Nevertheless, as neo-liberal reform develops, teachers need to participate in interactions with students, parents, school leaders, and colleagues, being immersed in a variety of complex and changeable emotional states for a long time. Emotional work and emotional labor are interwoven in the process of teaching [40].

There are various expectations and requirements of teachers' work, including the principles of emotion expression [59]. For instance, influenced by classic professionalism,

teachers are encouraged not to express much affectivity, whether excitement or anger. Like many other social service industries, controlling emotions is considered one of the manifestations of "being professional", so that teachers are supposed to be rational or neutral [40]. While under the call for an ethics of caring, teachers are expected to be sensitive to students' needs and talents and to be able to empathize, comfort, and motivate students by expressing or suppressing emotions in a timely and appropriate manner [60]. Moreover, teachers are expected to display and convey positive emotions in teaching [20]. The principle of sustaining passion requires teachers to exercise the aesthetic monitoring of themselves [61].

Teachers' emotional labor and their emotional rules for work reflect not only the social structure's constraints on the role of the teacher but is also a reflection of the agency and construction of the teacher's professional self [20]. Teachers are able to show emotional reaction flexibly and actively to the emotional rules within discourses, instead of strictly following the emotional rules [26]. In her research, Benesch (2020) found that teachers developed multiple ways to release and adjust to the stress or discomfort from the emotional demands of teaching-as-caring [62]. What is more, on the basis of a three-year genealogy research project on a teacher named Catherine, Zembylas (2005) found that emotional rules in teaching are historically contingent and that the teacher's self plays a part in emotional control, suggesting teachers are able to negotiate with the emotional discourses and to form their own emotional rules for teaching [40].

### 2.3. Special-Post Teachers in the Context of Rural China

The SPTRP was published by the Ministry of Education in China in 2006, intending to supplement the urgent shortage of teachers in basic education of the most impoverished regions in western China [31]. Later, it was gradually extended to the rural areas in the central and eastern parts of the country [63]. According to the policy, graduates who attend to the SPTRP and apply for special-post teacher positions must have received full-time higher education degrees or graduate from normal colleges. Special-post teachers are required to serve in the appointed schools by contract. The Chinese government would give special-post teachers additional welfare subsidies every year during their service period. After 3–5 years of teaching service, special-post teachers could choose whether they want to stay in rural schools to continue teaching [31]. If the special-post teachers have no intention to retain, they may transfer to schools in town centers or leave the teaching profession. The SPTRP has been implemented for more than fifteen years in China. Its implementation has increased the number of teachers in rural areas significantly and has optimized the educational level, age structure, and subject structure of rural teachers [64].

Nevertheless, it is still quite difficult to retain novice teachers in rural schools through the SPTRP. First of first, many young graduates choose to join the SPTRP, not out of passion for teaching, but instead they become special-post teachers simply because they were not able to find a better job temporarily or hope to optimize their career experiences by virtue of the SPTRP [65]. Once they receive opportunities of other occupations in urban or metropolitan areas, they are very likely to leave the rural schools after their service period. Hence, the special-post teacher position is often nicknamed the "occupational springboard" [66]. Secondly, the heavy workload and under-supported working environment in rural areas significantly undermine the early career development of special-post teachers. It is quite challenging for special-post teachers, who are young graduates and just beginning their teaching careers, to become accustomed to various difficulties in the impoverished areas such as excessive workloads, poor living conditions, lack of resources for supporting teacher professional development, and isolation from the cities [36]. Most importantly, many special-post students cannot identify with being teachers in rural schools. More and more urban-educated and urban-born college graduates become special-post teachers and teach in rural areas and have difficulty in adapting and integrating into rural schools. They experience confusion of identity, scarcity of self-efficacy, and little sense of belonging [67].

Special-post teachers have been found to have complex emotional experiences and diverse emotional labor at the workplace. For instance, Zhong and Zhang's case study showed that special-post teachers experienced extreme pressure due to the value gap between urban and rural education, unpleasant interactions with students and parents, work overload, and lack of sufficient resources for professional development [68]. Xiao and Wu's (2018) study narrated the "dilemma of leaving or staying" of five special-post teachers, who were filled with doubt, confusion, pain, helplessness, and sadness [69]. A handful of Chinese scholars explored how emotion changed in and interacted with the identity forming of special-post teachers in recent years [67,70]. Yet, there is still much unknown about what emotional struggles special-post teachers have to suffer and how they response emotionally and practically. This study aimed to explore special-post teachers' emotional labor from a post-structural perspective, focusing on how the emotions of special-post teachers are shaped by the rural school education discourse, and especially how agency is embedded in special-post teachers' emotional labor. Two research questions guided this study, as follows:

(1) What emotional conflicts or tension did special-post teachers face in the workplace?
(2) How did special-post teachers respond agentively to emotional conflicts or tension?

## 3. Methodology

This study drew on the qualitative data of an ethnographic case study conducted at a rural primary school located in Northern China. Qualitative research methods allow researchers to understand the inner experience of research participants and of grasping how meaning is formed through and within cultures [71]. It is quite useful for studying complex phenomena in specific contexts and is suitable for emotional research, since emotions are private, abstract, complex, and changeable. Qualitative research helps in understanding the context and process of special-post teachers' emotional production and development. More importantly, it can more truly narrate special-post teachers' voices and self-cognition on rural teaching practice. Ethnographic case studies are case studies "employing ethnographic methods and focused on building arguments about cultural, group, or community formation or examining other sociocultural phenomena" [72], combining the features of ethnography and the case study. The case study is one of the most commonly used approaches of qualitative research, referring to empirical inquiries that "investigate a contemporary phenomenon within its real-life context, especially when the boundaries between phenomenon and context are not clearly evident" [73]. Ethnographic case studies allow researchers to explore complex social phenomena with limited research resources, such as time and finances [33]. Considering that this research was expected to determine how the emotions of special-post teachers are interwoven and interacted with discourse in their workplace, an ethnographic case study approach was utilized in this study.

### 3.1. Research Context and Participants

The research site in this study is a rural primary school named the Big Tree Primary School. We chose this school as the research site for following reasons. First, the Big Tree Primary School is a six-year public school located in the countryside away from the suburbs of Beijing, which is a difficult-to-staff rural school with special-post teacher positions. Due to the schools in the district generally lacking young high-quality teachers, the Beijing Municipal Education Commission has set up special-post teacher positions in primary and secondary schools in the region since 2018, recruiting graduates from colleges and universities. The special-post service period is 5 years. Second, the samples from the Big Tree Primary School were adequate and abundant. There were more than ten special-post teachers with 0–5 years of teaching experience at the Big Tree Primary School, most of whom were willing to participate in the research. Last, but not least, limited by the COVID-19 prevention and control policies in mainland China, a mass of primary and secondary schools banned outside-school personnel. The Big Tree Primary School was one of the few schools that allowed academic researchers to do fieldwork. Furthermore, the school

location is relatively convenient for us. Therefore, considering the limited time and the operability of research, this school was chosen.

In order to collect more abundant data and to answer the research questions as completely as possible [74], purposive sampling and snowball sampling were utilized in this study. We developed three criteria for case selection in this study: 1. joining in the SPTRP; 2. being willing to participate throughout the whole research; 3. being employed for no more than three years. Since early career teachers might experience stronger emotional fluctuations [75], they are therefore more typical emotional laborers. Through the introduction of Mr. T, the Director of Teaching Affairs of the Big Tree Primary School, we met and established partnerships with two special-post teachers who were eligible for study in the teachers' office of Grade Five. Then, we constantly contacted other special-post teachers teaching in other grades by virtue of the two special-post teachers. Finally, we filtered ten research participants of different genders and education backgrounds that met the aforementioned criteria from all the special-post teachers that we contacted at the Big Tree Primary School. Both head teachers and subject teachers were involved. Table 1 below presents their information.

**Table 1.** Information of the ten participants.

| No. | Address | Gender | Taught Subjects | Being Head Teacher or Not | Teaching Experience | Profession | Education Background |
|-----|---------|--------|-----------------|---------------------------|---------------------|------------|----------------------|
| T1 | Ms. S | F | Chinese and Math | Yes | Two years and a half | Education | Master's Degree |
| T2 | Ms. Q | F | Chinese and Math | Yes | Two years and a half | Physics | Master's Degree |
| T3 | Ms. W | F | Chinese and Math | Yes | Less than one year | Chemistry | Master's Degree |
| T4 | Ms. O | F | Chinese and Math | Yes | Two years and a half | Chinese as a Foreign Language | Master's Degree |
| T5 | Ms. X | F | Chinese and Math | Yes | Two years and a half | Math | Master's Degree |
| T6 | Ms. C | F | English | No | Two years and a half | English Language and Literature | Master's Degree |
| T7 | Ms. P | F | English | No | One year and a half | English Translation | Master's Degree |
| T8 | Ms. Y | F | Art | No | Two years and a half | Designing | Bachelor's Degree |
| T9 | Mr. J | M | P.E. | No | Less than one year | Kinesiology | Bachelor's Degree |
| T10 | Ms. Z | F | Music | No | Two years and a half | Folk Instruments Performance | Master's Degree |

*3.2. Data Collection and Analysis*

The data collection for this study was conducted from December 2020 to March 2021. The specific data type we collected included participant observation, semi-structured interview, documents, pictures, and field notes. Table 2 shows the details of different types of data. In the first two months, we conducted the fieldwork. We asked Mr. T for a fixed seat in the teachers' office of Grade Five and were allowed to enter the school at any time on weekdays so that we might not only learn about the working conditions and school culture of the Big Tree Primary School but also communicate more with the teachers there. To integrate into the teacher community as quickly as possible, we went to the school at least three days a week, and every time we would stay there all day. At noon, we ate lunch with the teachers to have more informal and personal conversations with them, trying to form friendly relationships of mutual trust. We participated in teacher activities at the school, such as the teacher race walk competition, to befriend more teachers. We also visited the classrooms to talk with students and observed how special-post teachers interacted with students in order to obtain more information about special-post teachers' teaching practices, from which we left fieldnotes and physical materials. Through observation, we could catch more details about special-post teachers so that we might obtain more first-hand emotional episodes in authentic circumstances [76].

**Table 2.** Details of data collection.

| Data Type | Details |
|---|---|
| Semi-structured interview transcripts | 292,401 characters |
| Documents | 1 copy of the head teacher journal<br>3 copies of the "activity week" plans<br>2 copies of Beijing SPTRP documents<br>1 copy of a head teacher's timetable |
| Observation records | 2 sets |
| Pictures | approximately 50 MB |
| Field notes | 6184 characters |

A semi-structured interview is a kind of interview that seeks to obtain a description of the respondent's life world to explain the meaning of a described phenomenon [77]. Two rounds of semi-structured interviews were conducted in this study. The first-round was conducted offline from December 2020 to January 2021; we talked with teachers face-to-face. The second-round interview was conducted online in March 2021. Due to the lockdown policy for COVID-19, we were not able to visit the school. Thus, we interviewed the teachers online though the Tencent meeting platform. In the first-round of interviews, special-post teachers were asked about their education experiences, motivations of becoming special-post teachers, and their daily routines at the Big Tree Primary School. In the second-round of interviews, teachers were requested to talk about their feelings on interpersonal relationships and teaching practices in the workplace, especially their impressive stories and emotional episodes in teaching experiences [78,79]. Each interview ranged from 15 to 90 min. The main interview questions in our study for special-post teachers included the following: *(1) What at the workplace often affects your emotions? (2) Please recall 5 stories of anger, disappointment, grievances, helplessness, or sadness at school since joining the job? (3) Do you like working here? Why? (4) How would you deal with your emotions at work? Why? (5) Will you continue teaching at the Big Tree Primary School after the service period? Why?* In addition to the ten special-post teachers, other teachers at the Big Tree Primary School were also interviewed, including experienced teachers and an administrative leader. Table 3 below presents their information.

**Table 3.** Information of interviewees.

| No. | Address | Teaching Experience | Brief Introduction |
|---|---|---|---|
| T11 | Ms. G | More than six years | Head teacher |
| T12 | Ms. U | About thirty years | Experienced teacher; lesson preparation leader of Grade Five |
| T13 | Ms. K | More than seven years | Head teacher |
| T14 | Mr. L | Eight years | Science teacher; used to be a head teacher |
| A1 | Mr. T | About twenty-three years | Director of Teaching Affairs |

Notably, prior to data collection, informed consent forms were presented to the teachers who accepted our invitations. The informed consent form stated the purpose of this study, as well as how we hoped participants would help us and what rights they had once they joined in the study. Aspects of data collection were carried out with the permission of the research participants. That is, teachers participated in this study all joined voluntarily. We were committed to anonymizing information about the research participants in any text presentation. Therefore, the names of the school and teachers in this study are pseudonyms.

In this study, data analysis and data collection were carried out almost simultaneously since this avoided the busy situation of analyzing large amounts of data, while allowing researchers to continually reflect and improve upon the data collection process [80]. The qualitative data were analyzed by coding [77]. Transcripts and field notes were coded using descriptions in MAXQDA, and then the codes were repeatedly compared, generalized, and categorized. During the coding process, analytical memos were constantly recorded

to explore updated data and observation. Moreover, different types of data were utilized for comparison so that we could achieve triangulation from different sources of data to improve the validity of this research [81].

## 4. Results

*4.1. Special-Post Teachers' Emotional Conflicts at Work*

By and large, the analysis revealed four categories of emotional conflicts or tensions that special-post teachers faced and struggled with, which were embedded in various discourses of institution, education policy, and culture in the context of rural China.

### 4.1.1. Bored but Obligated to Do Miscellaneous Non-Teaching Tasks

For special-post teachers, doing non-teaching tasks is quite an annoying part of daily work, in particular for those who are also head teachers. At the Big Tree Primary School, every head teacher is required to manage the affairs of one stable class, including the construction of classroom culture, home-school cooperation, and the moral education of students. At the same time, a head teacher holds a concurrent post of Chinese and mathematics teacher in his or her class. That is, head teachers are all the teachers who teach Chinese and mathematics. In addition, head teachers also have responsibility for many non-teaching tasks almost every day, including administrative tasks, paper work, and social service, most of which are required by the school leaders and the government educational administration. For subject teachers, they also need to conduct non-teaching tasks, although not as much as head teachers do. The special-post teachers in this study claimed that doing miscellaneous non-teaching tasks was "quite disgusting".

> Head teachers have more cluttered things to do than subject teachers. These things have little relevance to your teaching, like having class meeting, doing copywriting, or having political studying. Sometimes you receive an administrative document unexpectedly in the WeChat group, maybe from the District Education Bureau, then you have to spend time preparing materials. Much of your time will be taken up, which you could have spent on resting or caring for your students . . . I hate doing these things. (S1-T1-S-20210309)

> Organizing files and writing some documents . . . (they) are not related to your teaching. The tasks are not so difficult, and they would not take your much time. But . . . it's just disgusting. (S1-T8-Y-20210124)

Although the special-post teachers in our study were unwilling to perform miscellaneous non-teaching tasks, they admitted that they had obligations to undertake and accomplish the tasks required by school and the government education administration. In the education management system of mainland China, rural school education and teacher workload are managed collectively. Rural schools, such as the Big Tree Primary School, have no autonomy in terms of teachers, funds, and decision-making, and rural school management is subordinate to the education management department of the local government [82]. Public school teachers are incorporated into the country's power system from various aspects, such as qualification certification, appointment and dismissal, the teaching process, and teaching results [83]. Therefore, our participants were unlikely to reject doing the non-teaching tasks from the government education administration or required by school leaders, even though they were reluctant to handle the tasks. Moreover, "head teacher of a class" is not only an institutional role setting but is also established by national educational policy documents. The head teacher system in primary and secondary schools is a significant part of the school system in Chinese basic education [84]. Hence, a head teacher is required to undertake all the affairs about the class he or she is in charge of.

> If the school leaders ask you to do it, do you have any other choice except obeying? You have to do it. It is frustrating, but it can't be helped. After all, many tasks are demanded by the government. Dare you say no? (S1-T1-S-20210309)

> Doing the non-teaching tasks about the class is part of a head teacher's job to some extent. You are unlikely to avoid it. (S1-T4-O-20201229)

### 4.1.2. Torn on Building a Traditional or Constructivist Classroom

Chinese educational reforms in recent decades have aimed to change the traditional national curriculum value into a constructive and practical one. A new ideal partnership for equality, negotiation, and dialogue among school personnel was advocated and has received unprecedented attention [85], having a deep impact on every aspect of the educational practice of China, including teachers' values on pedagogies and teaching modes [86]. In this study, special-post teachers mentioned that they were often torn on establishing a traditional quiet classroom within a hierarchical teacher–student relationship or a constructivist interactive classroom with an equal relationship with students. In their view, good teaching should be fun and passionate, encouraging expressions and interactions. They hoped to have friendly communication with students in and out of the class, filled with positive emotions.

> My ideal teaching is that we could teach and learn in a happy class climate. For instance, I can offer them a lot of learning resources and materials, the children have fun learning . . . I don't want to be a strict teacher. I hope they know that I want to be close to them. (S1-T3-W-20210120)

> I don't want students to hide when seeing me or to be afraid of me. I try to be their friend and supporter instead of a harsh controller . . . (S1-T5-X-20210120)

However, at the Big Tree Primary School, teachers are expected to build authority to make students fear them so that students would obey the discipline. Some experienced and elder teachers also suggested that young special-post teachers be harsh to students and keep the traditional one-way-teaching mode in the class to make sure students "listen to your words" and "study well", which confused special-post teachers.

> When I came to this school, everyone's request to me was: "Your class should be quiet first. Be strict to your students." I was criticized once for my class was too noisy by the old teachers. I felt so confused: "If you want children to innovate, if you encourage them to express, why make them close their mouths? Why let them fear teachers?" (S1-T4-O-20201229)

Moreover, special-post teachers also found that it was difficult to build an ideal constructivist classroom since they could not actually bear "an equal teacher–student relationship" they looked forward to. They might feel they are not being respected and get angry when they lose control of the class because of naughty students.

> If I were too close to them, they would treat me truly as their peers. Playing jokes on you, talking casually in your class, or not do their homework and tell you they do, which would absolutely annoy you. (S1-T1-S-20201208)

> I wish my students to be active and happy in my class. I mean, I hope they don't feel too constrained. But if some naughty boys are talking and laughing while I am talking, I would get angry and feel that I was not respected. (S1-T9-J-20210310)

### 4.1.3. Tangled about Academic Performance Coming First or Not

An inevitable conundrum for special post-teachers is the conflict between test-oriented culture and students' poor academic achievements due to insufficient sociocultural capital. Influenced by the Global Education Reform Movement, the basic education curriculum reform in mainland China started at the beginning of the new century was also a neo-liberal education reform. The competition-centered neo-liberal educational reform discourse utilizes performance as the key to evaluating educational quality [87]. Teachers who teach students productively in a technical sense are regarded as morally good teachers [88], which is distinguished from caring ethics, which emphasizes that teachers are supposed to focus on the specific needs of individual students. The contradiction between "academic

performance comes first" and "students' individual needs comes first" looks particularly prominent in the context of the current change in Chinese education [89]. At the Big Tree Primary School, academic achievement is utilized as an important standard to assess teachers. The participants also reported that poor student academic achievements would bring them frustration.

> Although the test performances of students are not emphasized in the daily work, at the end of a term, students' grades and rankings would be utilized as part of the evaluation criteria for we teachers. Poor student performance will be detrimental to teacher assessment and promotion. (S1-T1-S-20210309)

> Of course, if the average score of students in our class is lower than that of other classes, I would be upset. It is just like I cannot teach well, or I am a worse teacher than other teachers. (S1-T2-Q-20210117)

However, it is more difficult for rural students to adapt to the "urban-oriented" school culture and the new curriculum [90]. Compared with students in urban families, the cultural capital that these rural students have is more distinguished from formal school education, so it is difficult for them to gain academic achievements [91]. Students at the Big Tree Primary School do not have excellent academic performances in general. Teachers indicated that many of students did not have sufficient support in their studies and even had insufficient care in daily life. Some students come from single-parent families, or some of their parents are busy making a living or caring for another child. We noticed some children were wearing dirty clothes and shabby shoes. Thus, emotional labor was generated when special-post teachers were trapped in the tension between teaching practice and test-oriented culture. On the one hand, students' poor performance on tests and exams indicates the low-effectiveness teaching of teachers, bringing special-post teachers a low sense of achievement and frustration. On the other hand, special-post teachers also feel sorry that students do not have supportive cultural capital and enough care for their learning and development.

> After the math test, some of them only got 40 or 50 scores! I was so angry and depressed about their grades. You have to admit that most of these students are not similar to the children in the cities who have wealthy and supportive families or whose parents are well educated. I know it is not the children's problem because they need more specific teaching and more care. It is useless to blame them. (S1-T4-O-20201228)

### 4.1.4. Difficulty in Meeting Various Demands of Parents

It is quite a challenge for special-post teachers to cope with various requirements of parents. Interaction with parents is one of the most common forms of social interchange for teachers [92]. Influenced by the marketization of education policy in the reform, schools and teachers are encouraged to provide better educational services to parents who are educational consumers [93], which has deeply affected the relationship between parents and schools. At the Big Tree Primary School, teachers are required to maintain close and frequent communication with parents about students, trying to satisfy parents' expectations and requirements and to avoid conflicts with parents as much as possible. Special-post teachers also told us that they stayed cautious and even felt scared about being punished for parents complaining.

> Our school leaders warned us several times, emphasizing that teachers must pay attention to our ways of speaking with parents. We have to please parents as much as possible, in case of complaints to the District Education Bureau or the press by parents and teachers would be punished . . . I admit that I am afraid of parents. (S1-T1-S-20201217)

> I feel nervous when facing parents. I am afraid that parents have prejudice against me. There is a lot of news online now about parents complaining about teachers. Didn't you see that? (S1-T3-W-20210120)

Growing parental involvement has a great impact on the classical professionalism of teaching. It is more difficult for current teachers to defend and distance their expertise and to keep absolute authority from traditional "silent parents" [94]. Our participants reported that their interactions with parents were often filled with negative emotions because most parents here had distinguished values on education and teacher–parent relationships, requiring a lot of teachers' work.

> The parent asked me, "Can you leave more homework for the students? The homework you leave is now not enough. My child finishes homework before seven o'clock at night, and then he would play by himself. You should increase the difficulty and quantity of homework." But why should I increase the whole class's homework to fulfil his child's needs? A parent should not meddle in my homework setting. (S1-T5-X-20210120)

> One day, a parent told me that her child's shoes were slightly thin, so she hoped I would watch her child and did not let him go out to play. It was incomprehensible! I am not her babysitter or nanny! Why am I supposed to look after your child merely? Do I have nothing else to do? If she truly cared about the child's shoes, she should send a pair of warm shoes to school herself. (S1-T2-Q-20210117)

According to some experienced teachers, most of the parents are not very educated. Therefore, for special-post teachers, parents have no right to "play God" in teachers' teaching practice, and they are not supposed to shirk parental responsibility to teachers.

### 4.2. Special-Post Teachers' Agentive Responses Emotionally and Practically

The findings revealed that the special-post teacher in our study formed four types of agentive responses to the aforementioned emotional conflicts, in which process the intensity and frequency of emotional fluctuations of these young head teachers in this study have time-varying decreases. Our participants tried to engage in constant psychological and behavioral reflection and adjustment, which also involved their identity construction.

### 4.2.1. Resisting Given Tasks and Requirements by Minimizing Engagement

Although as head teachers our participants are unlikely to reject the obligation of non-teaching tasks required by governmental, educational administration, they tried to minimize their time and emotional engagement in such "boring" work by different tricks and methods, leaving themselves more space for professional work and having a rest. For instance, they would take turns taking part in the teacher training, which they thought "useless", and doing the minimum paperwork required by the governmental educational administration instead of full participation. They downloaded and copied required paperwork to cope with inspection by superior education administrative departments so that they would not spend much time writing by themselves. We also found that they taught a few excellent students who were capable enough to help them deal with easy tasks.

> When I turned around, a little girl was sitting beside Ms. S's desk, copying something from the computer screen. I leaned over to see, the girl was writing on a head teacher's week log (it is Ms. S's log), while the teacher's QQ showed a photo of another head teacher's week log . . . Ms. S told me that they were notified by the Education Bureau that the school would check head teachers' logs. She has to make up her log because she has not written it for a few weeks. The content of this kid of paperwork is not specifically inspected, as long as it is available. So, she just let students help copy it from another head teachers who have written logs. "The girl has good handwriting . . . I don't want to spend time writing this kind of boring and meaningless thing." And I saw another head teacher let her student write the head teacher's week log, too. (FN-20201218)

Moreover, when facing parents who wished the special-post teachers would do more jobs to "take care of" their own children, they would reduce their engagement on the basis of not conflicting with parents. In other words, our participants conducted their resistance agentively and smartly to these "additional" demands from parents that they thought did not belong to a teacher's duty.

> When parents wanted me to take more responsibility to their children, I would only verbally agree, but not truly make an effort to do it. For example, if parents told me to give more time to their children, such as "please give my boy more homework" or "the weather has been very hot recently, please take care of my child during the breaks and do not let him play outside the classroom", I would say yes on the surface, but I would not truly do that. I have no time to care about every child! (S1-T2-Q-20210117)

4.2.2. Taking Advantage of Emotions for Achieving Goals

The complex teaching practice at the Big Tree Primary School trapped special-post teachers in the emotional tension of various values. However, our participants explored their own ways of making use of emotional contexts and specifically of achieving their goals in teaching. Taking advantage of emotions helps special-post teachers achieve their educational purposes but also benefits themselves by working better. For instance, they tried to combine the features of traditional and constructivist teaching and established "a limited equal relationship" with students by utilizing emotions flexibly and contextually. This finding is similar to Yin and Lee (2012)'s findings that Chinese teachers attempted to instrumentalize their emotions to achieve teaching goals [20].

> Now I think, they are supposed to fear me in my class because I am their teacher who takes responsibility for their academic performances; they must listen to me and obey my words while studying. However, when class over, I hope they don't fear me so that they would communicate with me more about their needs . . . So, in the class, I am the teacher who controls. After the class, we might be friends. (S1-T2-Q-20210314)

> It is not good to show your anger to children. However, I still leave myself some space to lose my temper in order to let my students know that they have done something wrong, then let them reflect: "How should I correct my wrong behavior?" (S1-T4-O-20210309)

In the interaction with parents, special-post teachers learned to overcome their fear and caution of parents and communicated with parents more euphemistically and softly. They believed that managing emotional expression could be trusted by parents, which could be beneficial for gaining more cooperation from parents on student studying. This indicates that Chinese expression tradition would influence special-post teachers' emotional expressions, because in Chinese traditions it is considered immature to be unable to restrain emotions. Furthermore, if one has open conflicts with others, he or she would not only experience anxiety but also has to pay the interpersonal and survival costs [95].

> I discovered that there was no need to please parents too much. The more you are afraid of parents, the more parents will be afraid of you. You must express your opinions euphemistically to parents, not only to say what the parents like to hear but also to objectively express the shortcomings of their children. So that they might trust you more. (S1-T5-X-20210315)

> I used to be terrified of parents. However, it was useless to fear . . . When communicating with parents now, I am confirmed inside because I am supposed to have confidence that I am absolutely more professional than these parents. You don't need to please them. However, your tone should be empathic and gentle. You need a soft mouth to communicate so that parents would like to accept your words and become more cooperative. (S1-T4-O-20210309)

### 4.2.3. "Standing on the Other Side" Reflectively and Contextually

The special-post teachers in this study engaged in self-reflection by "standing on the other side" in the interactions with students, parents, and colleagues, for example, trying to consider why students did not learn effectively or perform ideally, why parents disagreed with them and "seldom cooperated with" them, or why older teachers at this school preferred traditional teaching. Through reflection, it became easier for them to empathize with students and parents, and then their emotional stress and sense of conflict would decline. This agentive coping approach also has typical characteristics of deep acting. Deep acting is regarded as the result of "seeking a more comfortable space" free from emotional dissonance [96].

> After I scolded him for crying, I reflected on my ways of treating students. Was I too harsh to them? . . . If you tried to understand them on the children's side, you would calm down to carry on your work. Additionally, if you tried to understand the issues from the perspective of parents, you would find their disagreements with you might not be that unreasonable. (S1-T2-Q-20210117)

Moreover, by "standing on the other side", our participants not only decreased their own emotional consumption but also adjusted their teaching practices. For instance, they learned to lower their expectations of test performance on poor-performance students and changed their ways of treating them. They found it necessary to grasp parents' requirements and expectations on students to design teaching more specifically.

> Taking a boy in my class as an example, almost every time he fails in Chinese and math tests, he obtains a score of 20 or 30. I was so upset and angry about him. However, later, I gradually understood that it is no use to be anxious because the child might not fit the school curriculum and teaching. So, I changed my expectation and standards of him. I mean, I cannot require him to get an 80 or 90 score as how I require of other students . . . When I dictate Chinese phrases, he would not be required to write it out. I just let him copy the words because I know he cannot write it out by dictation. (S1-T5-X-20210120)

> For some parents, academic performance is the most important thing they care about. However, for other parents, they only want their children to be happy and healthy at school. So, teachers have to know that not every parent has the same requirements for school, then we treat children specifically. (S1-T1-S-20201217)

### 4.2.4. Self-Concern and Seeking the Boundary of Care

As experience increases, the special-post teachers were more familiar with the educational ecology in rural areas and gradually recognized there were indeed many structural domains that they could not change or resist. They turned to paying more attention to caring for themselves, aiming to protect themselves from burnout and possible hurt from such an "unfriendly" environment.

> I persuaded myself that it was not worthy hurting my health because of work issues. Bad emotions might cause breast diseases. (S1-T8-Y-20210124)

> I am not going to "burn" myself as a candle-like teacher. Teachers are always coming to teach with enthusiasm and a desire for good teaching. However, in work interactions, we found that we might be hurt by things outside. We have to find ways to survive safely. (S1-T1-S-20201217)

To release themselves from pressure, confusion, and helplessness at work, our participants claimed that they learned to redefine the scope of a teacher's responsibility, trying to reject borderless caring and responsibility for work.

> A teacher is not a saint. I am not a person who can change the fate of these rural students as the parents wished. Not all the students could go to college to obtain higher education. Some students must be eliminated by the examination

system. It's not unreasonable, right? What we are able to change is limited. (S1-T4-O-20201228)

> Now teaching is a service industry. Children are born and raised by parents. We teachers merely offer knowledge and ways of learning that parents and children need. Teachers are unlikely to undertake parental duty, at most the co-operators of parents currently . . . Don't be so serious about work. (S1-T2-Q-20210117)

Notably, except for one special-post teacher who had no idea of her career development, others mentioned that they would not teach at the Big Tree Primary School after service. With turnover intentions, they regarded working emotions as part of labor for payment, indicating a trend of alienation in their emotions. Here we can include the following interview testimony:

> I console myself that I am paid SPTRP allowance for bad moods here. After all, I would leave for an urban school. (S1-T4-O-20210309)

> Well, the salary and benefits for special-post teachers are good here. So why let making money unhappy? I will be leaving anyway. (S1-T1-S-20201217)

## 5. Discussion

The current study focuses on the emotional labor of Chinese special-post teachers from a post-structural perspective. It stresses that teacher emotion is shaped by the norms and rules within power relationships, and it highlights that teachers have agency to negotiate their identity with emotional discourses. The results showed four categories of emotional conflicts that special-post teachers face and four types of agentive responses that special-post teachers give to these emotional tensions. This discussion is guided by the two research questions.

### 5.1. Role Overload and Conflicts of Special-Post Teachers

Our findings indicate that Chinese special-post teachers struggle with multiple roles within complex discourses due to the urban–rural dual structure of Chinese society. In other words, the urban–rural dichotomy has intensified the role overload and conflicts of special-post teachers. Influenced by the Global Education Reform Movement, teachers have been assigned more requirements from policy makers, parents, and other social forces in this changing age [59]. The intensification of teacher workload has become a worldwide issue. In the context of rural China, teachers were found to be overworked due to a societal shortage of teachers, excessive administrative tasks from basic-level government, ineffective parental involvement, and lack of organizational support [97]. Scholars have pointed that the current Chinese rural teachers have multiple role conflicts because of the urban–rural dichotomy, social and public function changes of teachers, differences in teaching professionalism, and national–local knowledge gaps [98]. Moreover, many rural families shift their educational responsibilities to school [99], which might increase rural teachers' workloads and role burdens. Rural teachers are regarded as outlanders, public servants, service providers, reform practitioners, etc. [100]. Hence, Chinese rural teachers exactly experience role overload and conflicts from work. The role conflicts and identity dilemmas that special-post teachers have could be more prominent. Special-post teachers are regarded as "special role models" at rural schools and the hope of rural education reform and improvement, since they are the main force of college students and have received formal higher education in general. They are also actually marginal roles in the teacher community because they are not formal members of rural public schools during the service according to the SPTRP policy [101]. Special-post teachers are part of educational resources to narrow the gap between urban and rural education. Therefore, the roles that special-post teachers play are likely to be more inherently ambiguous and conflicting in the context of the urban-rural dual structure of Chinese society.

Administrative workloads that are less related to teachers' professional teaching might lead to teacher dissatisfaction. Scholars have claimed an urgent demand to reduce

teachers' administrative workloads, since teachers with greater administrative workloads are less likely to spend time on instructional teaching [102]. In the current study, the special-post teachers emphasized their displeasure and stress of doing administrative non-teaching tasks, and those who were also head teachers were obligated to do more, and they also had to experience more complex role conflicts. This finding suggests that the interference of executive discourse extremely affects the role stress of Chinese teachers, especially head teachers. The accountability within the hierarchical system could shape and reshape teachers' work and is the main source of teachers' administrative workloads [103]. Due to the lack of a modern school system set-up, Chinese schools have to undertake administrative tasks required by the government, which is highly related to teacher work overload [104]. As persons in charge of classes, head teachers spend a considerable amount of time on non-instructional work, with more than half spent on miscellaneous issues that are not related to teaching practice [105]. Being a head teacher is even seen as a "high-risk career" in rural China.

Similar to what Loh et al. (2016) found, the culture of performativity and grading stress brought emotional burden to special-post teachers [60]. These findings suggest that neo-liberal reform discourse represented by performance culture and constructivism were strongly resisted by local and traditional discourse and professional ethics of care. Evidence has also demonstrated that contemporary pedagogic discourse in rural areas is currently still rooted in traditional elements [106]. Influenced by Confucianism, ancient Chinese gave respect to the dignity of teachers. Teachers were regarded as the holder of knowledge. The status of teachers was based on supreme authority in culture and knowledge. There was a clear distinction and an obvious inequality of power relationship between teachers and students, which is different from the equal and cooperative teacher–student relationship that Chinese educational reform proposed. Moreover, in the reform, students' academic performance is considered a basis for the quality of teaching, which could be evaluated and kept accountable. As the marketization of education develops, teachers' professional authority is dissolved by the logic of consumption, and teachers' professionalism is simplified into performative skills. Teachers might give more attention to students who can obtain high scores and ignore poor-performing students [87], which is also definitely against the caring ethic of the teaching profession. Teachers are regarded as those who provide educational services. The nature of teachers' work has changed, and teaching has been embedded into the new accountability and performance systems in neo-liberal education reform. Yan (2019) also pointed out that rural teachers tended to become "knowledge servants" and "teaching workers" because of the development of teacher professionalism and the regulation of technocratic education reform [107].

*5.2. Emotional Reflexivity and Vulnerability in Special-Post Teachers' Identity Construction*

Teachers' responses to emotional discourses and emotional tensions are inseparable from their perspectives on themselves, that is, their identity. In the current study, our findings suggested that Chinese special-post teachers might regard themselves as educators, professional-service-providers, or passers-by in the emotional struggles of discourses. Teacher identity is quite complex, and a teacher might have more than one identity at the same time [108]. In our findings, the special-post teachers at the workplace value their teaching expertise while also attempting to resist non-teaching workloads and additional responsibilities. Notably, almost all of them regarded themselves as passers-by of the rural school and believed that they would eventually leave for better occupations. This is due to the SPTRP policy being an alternative hiring policy with exit channels to a great extent [33]. After all, the service period of special-post teachers is limited, and special-post teachers have the right to leave. The special-post teachers in this study minimize excessive consumption of passion and concern in their work to comfort themselves and teach students effectively.

Teachers are agents of working conditions and educational change [109], rather than puppets in teaching practice. Teacher agency explains how teachers "critically shape their

responses to problematic situations" [110]. As reflexive agents, teachers are influenced, but not determined by, social structures [111]. In this study, special-post teachers did not know how to cope with all these complex emotional discourses at the beginning. They experienced extremely strong emotional fluctuations in their first years. As their teaching experiences increased, they gradually formed their "own" ways to respond to and negotiate with the emotional discourses for work. Their negative emotional experiences and emotional burden were also reduced. These findings demonstrated that teachers in the current study had emotional reflexivity in their emotional experiences and identity construction. According to Holmes, emotional reflexivity "refers to the inter-subjective interpretation of one's own and others' emotions and how they are enacted". Emotional reflexivity could be understood as "the practices of altering one's life as a response to feelings, and to interpretations of one's own and others' feelings, about one's circumstances" [112]. A teacher's critical emotional reflexivity is "a process that is deeply embedded in social structures and school norm" and a "critical reflection on one's own emotions and assumptions as a teacher and learner" [113]. Individuals have to learn to self-regulate and become more dispassionate and measured, forced by the regimes of reflexivity and increasing social demands for emotional control. Through working experiences, special-post teachers learned to promote emotional understanding with students and parents. Emotion and agency were interplayed and intertwined in the process of the identity construction of teachers.

Special-post teachers made a compromise within emotional-consuming structures and complex discourses in rural areas, which might refer to their vulnerability. Vulnerability is generated from the expectations and norms of professional teaching realities that mediate or constitute teachers' senses and views of themselves as professionals [114]. Vulnerability is closely related to teachers' beliefs and commitments in both educational and social contexts according to school policies, institutional rules, and social norms [115]. In the current study, most special-post teachers in this study graduated from noneducational majors and did not have normal pre-service teacher education. Additionally, some of them admitted that they entered the Big Tree Primary School since they had not obtained better job opportunities. Therefore, their commitment to the teaching profession might be more fragile. This might increase their vulnerability when facing complex educational and sociocultural contexts in rural areas. Moreover, the unsupportive working conditions for teacher development in rural schools of China might lead to more vulnerability might not be supportive enough. Chinese cultural tradition may also trap special-post teachers in this paradox. Gao (2008) found that Chinese cultural tradition might add to teachers' vulnerability, since the tradition was used to give teachers extreme authority but also imposed burdens on teachers. Teachers who expressed complaints about their work would be thought moral failures by parents and educational administrators, since that is not in line with a selfless and altruistic image of teachers that is promoted, even though in Chinese culture, teaching is a respected profession [116]. Repressing negative emotional expressions is likely to increase young special-post teachers' sense of vulnerability for work.

## 6. Conclusions and Implications

This study aimed to explore the emotional labor of special-post teachers from a post-structuralist perspective. The results showed four categories of emotional conflicts that special-post teachers face. First, special-post teachers think doing non-teaching tasks is boring, but they are not able to reject them. Second, special-post teachers feel torn with their students on building a traditional or constructivist classroom. Third, special-post teachers experiences conflict between whether "academic performance comes first" or "students' individual needs come first". Fourth, the various demands from parents distress special-post teachers a lot. Then, four types of agentive responses that special-post teachers give to these emotional tensions were summarized. First, special-post teachers minimize their engagement in the tasks or requirements that they thought unrelated to them. Second, special-post teachers usually make use of emotions in order to achieve their goals for work. Third, special-post teachers often try to take another perspective

reflectively and contextually in interpersonal interactions. Last, special-post teachers also learn to give self-concern and to build a boundary of care for work, so that they may decrease their stress and avoid exhaustion. The findings highlight three sub-identities that Chinese special-post teachers might have: educator, professional-service-provider, and also passer-by. Special-post teachers' emotional labor was shaped and identities were formed in multiple discourses.

The following implications of this study are proposed: Theoretically, the current study explored the emotional labor of special-post teachers from a post-structuralist view, which provides Chinese rural experience with post-structuralist teacher emotion research since previous studies about post-structuralist teacher emotion generally focused on language teachers. The urban–rural dichotomy of economic society has become one of the keys to understanding teacher emotion in the context of remote or rural areas. Then, it supports that various discourses would not only control teachers but also offer resources for individual resistance and coping [40,42], as well as supports that emotion is a separable part of reflexivity [112]. Practically, this study first contributes to special-post teachers' mental health wellbeing. Our findings suggest that emotion is a significant dimension of special-post teacher retention. In addition to material support such as salary allowance, special-post teachers, especially novice special-post teachers, need more spiritual and emotional support from rural and school communities to decrease their alienation from the countryside. Our findings also support that it is necessary for local governments to reduce non-teaching administrative workloads for rural teachers. Second, our findings reveal that special-post teachers may not be familiar with the learning conditions and growth environments of rural students. Thus, it is necessary to improve local knowledge and culturally responsive pedagogy into the teacher training or learning activities for special-post teachers. Third, through the current study, we recommend that the entrance to the SPTRP should be improved. Considering the complex conditions in rural education, only college graduates with stronger educational commitments should be selected to participate in the program.

## 7. Limitations

The current study has several limitations. First, this study was finished in less than four months during the COVID-19 pandemic. Influenced by the pandemic prevention and control policy of China, we were not able to do field research in more rural schools to explore the factors beyond school level that might shape the emotional labor of special-post teachers. Furthermore, factors such as the diversity of local educational ecology, differences in policies of SPTRP in provinces, and imbalance in regional social economics of China would affect special-post teachers' living and working environments, which would then influence their opinions and emotional labor. Future research can collect data from distinguished schools or in more regions to understand special-post teachers within more multi-dimensional information. Second, although this study used interviews and other methods to collect data, given the private nature and variability of emotions, it is likely that many details were missed by the teachers interviewed because they did not have clear memories of the emotional episodes. Future research could use methods such as inviting participants to keep an emotion diary to record their emotions in as timely a manner as possible. Third, our findings suggested that special-post teachers are quite likely to have emotional reflexivity in their identity forming and negotiation. Yet, there is still much unknown about how special-post teachers' emotions drive their teaching practices or how their actions in teaching influence their emotional regulation. Future research might explore more about the interplay between special-post teachers' emotions and agency.

**Author Contributions:** Conceptualization, H.S.; Methodology, H.S.; Formal analysis, J.W.; Resources, H.S.; Data curation, J.W.; Writing—original draft, J.W.; Supervision, H.S.; Funding acquisition, H.S. All authors have read and agreed to the published version of the manuscript.

**Funding:** This work was supported by the National Natural Science Foundation of China (71974016); the project "How to prepare a good teacher" funded by the International Joint Research Project of Huiyan International College, Faculty of Education, Beijing Normal University (ICER201905); and the BNU First-Class education Discipline Plan (YLXKPY-XSDW202207).

**Institutional Review Board Statement:** Ethical review and approval were waived for this study, due to the unavailability of the Institutional Review Board (or Ethics Committee) at the affiliation of the first author who was responsible for data collection.

**Informed Consent Statement:** Informed consent was obtained from all subjects involved in the study.

**Conflicts of Interest:** The authors declare no conflict of interest.

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
