# Peer review of "“Learn to Conserve Your Passion and Care”: Exploring the Emotional Labor of Special-Post Teachers in Rural China"

_sustainability, doi:10.3390/su15031991_

Round 1

Reviewer 1 Report

First of all, thank you very much for giving me the opportunity to review the paper entitled: “Try not to take your work so seriously”: A case study of the emotional labor of special-post teachers in rural China - and some recommendations and suggestions are provided:

I would like to start with the first question regarding the following manuscript, 

1-    What is your research contribution?

2-    What does it mean special-post teacher positions? What’s the difference between the following teachers and other types of teachers? Readers would like to understand the difference and why they are called by special-post teachers positions and how they are recruited (sustainability is an international journal that attracts readers from over the world and we don’t understand how the educative system in china is. Do they have to take special courses or training…? More details are needed.

Abstract

An abstract should include the following sections: background/motivation, aim, methodology, principal findings, and conclusion/significance.

Please try to organize the abstract according to the following.

The method in the abstract is not clear, the findings and conclusion/significance.

Introduction

Line 34-38:

Scholars found that working conditions and school characteristics, such as a low salary, a heavy workload, a lack of professional development resources, insufficient administrative support and the race and gender of the teacher might be reasons for rural teacher turnover (Mason & Matas, 2015; 37 Carver-Thomas & Darling-Hammond, 2017; Nguyen, 2020). 

It is better to refer to scholars in the area of the study (Chinese context).

Literature review

Please refer to new literature, most of the references are old.

Line 172: this line needs to be improved 

Line 174-175: this sentence is not complete “After 3-5 years of service, special-post teachers can choose whether they want to stay in rural schools to continue teaching.”

The SPTRP has been implemented for more than fifteen yearsmin China. 

Line 263: what do you mean by teaching qualification? Is it special training?

Background must include very current references (2022, 2021)

Methodology

This section needs extensive revision for meaningful understanding. Please add information in a sequence, making it easier for the reader to understand.

how this research was conducted and how many times when is it online or offline, the reason for observation, visiting the school, and making friends with teachers.

What was your research design? 

What was the population? How did you recruit samples? Which sampling did you use and why? Add more details about them.

Does the study was approved by the Ethics committee of the local university?, 

Do authors take informed consent from subjects involved in the study?

The Data Analysis section must be more explicit and must better argue the steps that follow and why they do so, the thresholds of results to consider of interest, the effect sizes or real significances in addition to the statistical significances.

Line 272: This school was established in this study?

Table 1: only female teachers participated in the study? 

Line 293: but also might allow us to establis rapport with our 

Also, …. Establish

Line 320-322

In March 2021, on the basis of existing data, another round of online interviews was started. Each interview ranged from 15-90 minutes, including face-to-face and online interviews. 

It was cited that interview in march was online and later face-to-face and online, which is confusing 

Conclusion

There’s no conclusion for the study,

Recommendation, Limitation, Implication

It is better to add recommendations, limitations, and Implications of the study.

The Discussion and conclusions must answer the research question or problems if the objective has been achieved or not, if the hypotheses are confirmed or not; and a theoretical interpretation in light of empirical evidence and current background (2022, 2021) reviewed in the Background. Limitations of the study should be identified in greater detail and future prospects or avenues for solutions.

References

Please double-check for the references, they are not following the journal recommendation, and some references and not well cited check if there are no mistakes in the references, for example, ref 4-5 

references (and DOIs https://doi.org/xxxxxxxx) of all references; and of those that do not have a DOI, the URL that allows the direct location of the same.

I hope the following comments and suggestions can help to improve the manuscript.

Good luck.

Author Response

Dear reviewer,

On behalf of my co-author, we are quite grateful for your positive and constructive comments on our manuscript (ID: sustainability-2017474).

We have studied your comments carefully and reply to all the comments which marked in red. We have tried our best to revise our manuscript according to the comments. 

Please see the attachment for details, which we would like to submit for your kind consideration.

Yours Faithfully,

Jianjian

Reviewer 2 Report

1. The title is gimmicky and misleads the reader - get to the point

2. There should be stronger definitions for the keywords in the study - like Emotional Labor

3. In the findings section the reader can be confused - sometimes extracts from interviews are not clearly presented - we are not sure if it is the author or participant. 

4. The paper must be thoroughly edited as there were numerous language problems detected.

5. There must be stronger conclusions backed by literature (especially in research linked to Novice Teacher problems which lead to them quitting their jobs.

Author Response

(The authors gave the same response as above.)

Reviewer 3 Report

The study investigated an interesting topic ““Try not to take your work so seriously”: A case study of the 2 emotional labor of special-post teachers in rural China” which might be of interest to the readership of Sustainability. However, I am concerned about several issues, which need to be addressed before the manuscript can be considered further.

Firstly, the title does not really reflect the main content and contribution of the paper. It needs to be revised.

The introduction and literature need to be more focused to identify the importance for your current study.

The concept of ‘special-post teachers’ need to be much clearer. This is particularly important for the readers outside China.

The findings are quite interesting, but they need to be reorganized to enhance readability.

Regarding the discussion, there is a need for a more in-depth discussion in relation to the updated literature to highlight the knowledge contribution of your study. The authors could consider this meta-analysis on teachers’ stress and anxiety in their literature and discussion: Ma, K., Liang, L., Chutiyami, M., Nicoll, S., Khaerudin, T., & Ha, X. V. (2022). COVID-19 pandemic-related anxiety, stress, and depression among teachers: A systematic review and meta-analysis. Work, 1-25.

I hope my comments are useful to the authors in their revisions.

Good luck!

Author Response

(The authors gave the same response as above.)

Round 2

Reviewer 1 Report

Thanks to the authors for revising the manuscript.

the manuscript can be accepted for publication

Author Response

Dear reviewer,

On behalf of my co-authors, we deeply thank you for giving us an opportunity to revise our manuscript again.

We have learned your comments very carefully. Since the issue of poor readability was mentioned several times, we again asked a native speaker to do a proof-reading. Attached please find the revised version, which we would like to submit for your kind consideration.

We would like to express our great appreciation to you for comments on our paper. Thank you so much and best regards.

Yours sincerely,

Jianjian 

Reviewer 2 Report

Much improved version. BUT needs further proof-reading. Language needs to be improved.

Author Response

(The authors gave the same response as above.)

Reviewer 3 Report

Thank you for considering my comments and revising the manuscript. It has now been ready for publication.

Author Response

(The authors gave the same response as above.)
